# Dispersal Behavior Characters of *Spodoptera frugiperda* Larvae

**DOI:** 10.3390/insects14060488

**Published:** 2023-05-24

**Authors:** Yong-Ping Li, Su-Yi Yao, Dan Feng, Robert A. Haack, Yang Yang, Jia-Lan Hou, Hui Ye

**Affiliations:** 1School of Agriculture, Yunnan University, Kunming 650500, China; 2Institute of International Rivers and Eco-Security, Yunnan University, Kunming 650500, China; 3Yunnan Academy of Forestry and Grassland, Kunming 650201, China; 4USDA Forest Service, Northern Research Station, Lansing, MI 48190, USA; robert.haack@usda.gov; 5School of Biology, Yunnan University, Kunming 650500, China; 6School of Ecology and Environmental Sciences, Yunnan University, Kunming 650500, China

**Keywords:** *Spodoptera frugiperda*, dispersal, crawling, ballooning, airflow

## Abstract

**Simple Summary:**

The fall armyworm (FAW), *Spodoptera frugiperda*, is one of the world’s most important corn pests. FAW larval dispersal is a major factor contributing to local spread and damage by this pest. In this study, we investigated the crawling and ballooning dispersal characteristics of FAW larvae in the laboratory and found that airflow was an important factor influencing larval ballooning and, thereby, the long-distance dispersal of FAW larvae. These results provide scientific information for field monitoring and control strategies against FAW.

**Abstract:**

The fall armyworm (FAW), *Spodoptera frugiperda* (J. E. Smith) (Lepidoptera: Noctuidae), is a major pest of corn worldwide. FAW larval dispersal is an important life strategy that influences FAW population distribution in corn fields and subsequent plant damage. We studied FAW larval dispersal in the laboratory with sticky plates placed around the test plant and a unidirectional airflow source. Crawling and ballooning were the main dispersal means of FAW larvae both within and between corn plants. All larval instars (1st–6th) could disperse by crawling, with crawling being the only dispersal mechanism for 4th–6th instars. By crawling, FAW larvae could reach all aboveground parts of a corn plant as well as adjacent corn plants where leaves overlapped. Ballooning was used primarily by 1st-3rd instar larvae, and the proportion of these larvae that used ballooning decreased with age. Ballooning was largely governed by the larva’s interaction with airflow. Airflow influenced the direction and distance of larval ballooning. With an airflow speed of about 0.05 m/s, 1st instars could travel up to 196 cm from the test plant, indicating that long-distance FAW larval dispersal depends on ballooning. These results increase our understanding of FAW larval dispersal and provide scientific information for the development of FAW monitoring and control strategies.

## 1. Introduction

*Spodoptera frugiperda* (J. E. Smith) (Lepidotpera: Noctuidae), commonly known as fall armyworm (FAW), is one of the world’s important corn pests [1,2]. It is native to tropical and subtropical regions of the Americas, was first reported in Africa in 2016 [3,4], and was reported in Asia in 2018 [5,6]. As of 2023, it has been widely distributed in more than 80 countries in the Americas, Africa, Europe, Asia, and Australia, posing varying degrees of threat to local corn planting industries [7,8,9,10].

FAW have a high reproductive potential, with females often laying 2–11 egg masses, each containing 100–200 eggs, resulting in about 1500–2000 eggs laid during their lifetime [2,11,12]. The eggs are mostly laid on the leaves of corn plants. After hatching, the larvae disperse and mainly feed on leaves; they also bore into corn tassels, which causes damage to the ears of corn [7,13]. FAW larvae have six instars and can develop in 9–56 days depending on temperature [14]. After larval development is complete, an individual falls to the ground and pupates in the soil for about 7–37 days, depending on the temperature [15]. Adults fly to corn fields after emergence and begin to lay eggs on corn leaves 2–3 days after mating [11,16]. It takes about 30 days for the FAW to complete one generation in temperate regions [15,17]. As a result of FAW feeding on corn leaves and developing ears, corn yields decline, and at times there are complete crop failures when larval populations are high [1,2,18]. 

Larval dispersal is an important adaptive feature of FAW in response to their high reproductive potential, which aids them in maintaining population growth [6,18,19,20,21,22]. When newly hatched larvae are crowded, their fate will often depend on whether they can disperse quickly enough to find suitable food, reduce cannibalism, and avoid predation [12,23,24,25,26]. In a corn field, FAW larvae mainly appear in the upper, younger, and more nutritious tissues of the plant [6,27,28,29,30]. The younger leaf tissues and developing reproductive tissues of corn plants are more tender and nutritious, which is beneficial for the faster development of FAW [28,31]. FAW have the habit of cannibalism. FAW larvae aged 2–4 days began to exhibit cannibalism, while larvae aged 10–12 days showed high cannibalism rates [24,32]. About 23% of larvae die from cannibalism within 2 days after hatching, with a high rate of 40% within 4 days [32,33,34]. Dispersion of FAW larvae helps reduce cannibalism and ensure a higher survival rate [26,35].

Studies on FAW larval dispersal and spread are important for understanding the damage caused by FAW larvae at the field level, as well as for the development of monitoring, prevention, and control measures [2,10,23]. Although FAW larvae disperse by crawling and ballooning on silk threads has been observed, the degree to which these activities occur and how they are influenced by environmental conditions remains unclear [20]. Crawling enables larvae to move to any above-ground part of the corn plant, resulting in scattered distribution on the entire plant [2,6,27]. One study found that FAW larvae concentrated on young tissues and organs of the host plant [28]. The direction in which FAW larvae crawl after hatching determines which parts of the corn plant will likely suffer more damage. 

FAW larvae can secrete a silk thread and use it for dispersal in a process called ballooning [36]. FAW larvae can reach nearby or distant plants by ballooning [35,37]. Ballooning can expand the area of infestation around individual plants with egg masses, which can create an aggregated spatial distribution of the larvae in the field [35]. The ability to disperse by ballooning increases the invasiveness of FAW [38]. Therefore, it is important to understand how far ballooning migration can occur and what environmental factors affect dispersal in the fields. Some studies found that airflow affected the direction of FAW larval dispersal in field [35,37]. In one review [36], the authors stated that ballooning depended on the larva’s interaction with its environment (including temperature, humidity, airflow, vibration, and light level). However, due to the fluctuations of environmental factors in a typical field experiment, we attempted to control certain variables to illustrate more precisely how airflow affects FAW larvae dispersal.

The main objectives of this greenhouse study were to: (1) make detailed observations on the crawling and ballooning dispersal behavior of FAW larvae, and (2) record the effect of airflow on the percentage of FAW larvae that dispersed by ballooning as well as the distance travelled and the direction of dispersal. Our aim was to elucidate the dispersal behavior of FAW larvae and the factors that influence these behaviors under controlled conditions in hopes that such information would lead to improved monitoring and control strategies for FAW.

## 2. Materials and Methods

### 2.1. Plants and Insects

We planted individual corn plants (hybrid “Diwo 4”) in plastic pots (15 cm high, 18 cm diam) in the greenhouse of Yunnan University. The test plants used in the studies were about 90 cm tall with 13 leaves.

FAW larvae were collected from corn fields near the campus of Yunnan University, Kunming, Yunnan, China, and reared in the laboratory for two consecutive generations. The rearing conditions included room temperatures of 26 ± 1 °C, a relative humidity of 65 ± 5%, and a photoperiod of 12:12 [L:D] h. Based on a study by Li et al. [39], the larvae were fed an artificial diet consisting of 0.1% fine wheat bran, 0.1% soybean flour, 0.04% yeast extract, 0.03% casein, 0.03% agar powder, 0.003% vitamin C, 0.002% cholesterol, and 0.002% potassium sorbate. 

First- and second-instar larvae were reared in plastic boxes (245 × 160 × 45 mm) with the artificial diet replaced every 2 days. Third instars were transferred into 24- grid boxes (190 × 125 × 35 mm) where they were reared individually to prevent cannibalism. During development, thirty first-, third-, and sixth-instar larvae were weighed to analyze the relationship between larval weight and ballooning. Fully mature larvae were put in a plastic box containing sand, which they entered to pupate. Newly emerged adults were transferred to plastic boxes (245 × 165 × 90 mm) in pairs, where they mated, and then were placed in other boxes of similar size to oviposit. We used folded pieces of paper as an oviposition substrate and changed the paper every day. Later, we placed the paper sheets containing eggs in an incubator at 26 °C and made daily observations with a stereoscopic microscope (Olympus, Tokyo, Japan, SZ2-ILST) until darkening of the larval head capsules became visible, indicating that egg hatch would occur soon. The number of eggs in each mass were counted and these eggs were used in later experiments.

### 2.2. Larval Dispersal Observations

This experiment was conducted in a greenhouse on the campus of Yunnan University. The experimental arena consisted of a 3 m × 3 m area covered with 225 sticky boards, each measuring 20 cm × 20 cm (Figure 1). We calculated the percentage of 1st- to 3rd-instar larvae that dispersed by ballooning and crawling, the length of the ballooning silk thread, and the percentage of 1st instars that dispersed by ballooning for a second time. In addition, many of the 1st instars that dispersed by ballooning were collected soon after ballooning, placed back on a test plant, and observed again to determine how many would balloon a second time.

#### 2.2.1. The Percentage of 1st- to 3rd-Instar Larvae Ballooning and Crawling 

The corn plants were placed in the center of the sticky plates. Tests were conducted, first with 200 1st-instar larvae, next with 50 2nd-instar larvae, and then again with 10 3rd- instar larvae. The test insects were placed on the upper side of the 9th leaf of the corn plant. Two pieces of double-sided tape were attached to the stem of corn plant approximately 6 cm above and 6 cm below the attachment site on the 9th leaf, and were intended to capture larvae as they crawled past. We used the number of larvae captured on the tape to calculate what proportion of larvae that crawled upward or downward along the main stem. Given that Rojas et al. [40] reported that most larvae disperse from the original oviposition plant within 1 h after hatching, we recorded the number of larvae captured on the sticky plates, on the sticky tape, and on the stem after 1 h. These numbers were used to calculate the percentage of larvae that dispersed by ballooning and crawling. The experiment was repeated three times for each larval age class. 

#### 2.2.2. The Length of the Ballooning Silk Thread

Once the 1st instar larvae were placed on the test plants, they were monitored continuously for about 1 h in person. During this time, the length of the ballooning silk thread was recorded for 20 individuals. In the initial stages of ballooning, FAW larvae usually suspend themselves from a silk thread that is attached to a leaf. We used the distance between the leaf attachment site and the suspended larva as the length of the silk thread.

#### 2.2.3. The Percentage of Larvae Ballooning Twice

Several of the 1st-instar larvae that dispersed from the test plant by ballooning were collected, placed back on the 9th leaf of another test plant, and then observed for 1 h. The number of larvae that dispersed by ballooning a second time was recorded, and these values were used to calculate the percentage of larvae that ballooned twice. This study was replicated three times, using 100 larvae each time.

### 2.3. Impact of Airflow on Ballooning Behavior

As in experiment 2.2, we used the same 3 m × 3 m experimental arena with sticky traps on the floor and a clear PVC (polyvinyl chloride)-covered windbreak shelter that was constructed to surround the test arena on all four sides, with two opposing sides that could be opened to allow airflow (Figure 1). The walls were 2 m high, which was taller than the test corn plants. One corn plant was placed at the center of the arena with sticky plates around it. A paper sheet with about 300 eggs was stuck to the underside of the 9th leaf of the test corn plant, which was about 50 cm above the ground. Double-sided tape was applied to the corn stem, about 6 cm above and 6 cm below the point the leaf where the eggs were attached. These traps along the stem were used to capture any larvae that dispersed by crawling. Observations were made twice daily, at 9 a.m. and 5 p.m., for two consecutive days. We recorded the number of larvae on the sticky plates and the distance and direction of each captured larva relative to the test plant. In addition, the number of larvae captured on the sticky tape above and below the 9th leaf was recorded. After the 2-day test period, the egg sheet was viewed under a microscope to determine the number of hatched and unhatched eggs.

To investigate the effect of airflow on ballooning, testing was done both with and without unidirectional wind. The test arena was again used, and the wind-free treatment was produced by having all four sides of the shelter closed and no source of wind. For the treatment with wind, two opposing sides were opened, and a fan was used to provide steady airflow from west to east, with a speed of about 0.05 m/s. The same methods for egg and trap placement were used in both treatments. Once testing was complete, the percentage of larvae dispersing by ballooning and crawling was calculated. In addition, the location of each larva captured on a sticky plate was assigned to one of four directional quadrants [east (E), west (W), south (S), and north (N)] relative to the base of the test plant and the west-to-east airflow pattern.

The wind-free and wind treatments were each repeated three times. During the experiment, a HOBO (Mx2301A, Onset, Bourne, MA, USA) was used to record the temperature and humidity every hour.

### 2.4. Statistical Analysis

The *t*-test was used to compare the percentage of FAW larvae that crawled upward and downward, as well the average distance of dispersed larvae under wind-free and windy conditions. One-way ANOVA followed by Duncan’s multiple range tests (*p* = 0.05) were performed to compare the differences in the mean number of dispersing larvae in the different directions under wind-free and windy conditions. Before ANOVA, the data were analyzed for homogeneity of variance with Levene’s test. Because Levene’s test was significant, the data were square root-transformed first. An alpha level of 0.05 was used for significance. Statistical analyses were performed using SPSS Statistics 26.0 (IBM, Armonk, NY, USA). 

## 3. Results

### 3.1. Larval Dispersal Behavior 

We observed two forms of silk-thread dispersal. In the first case, larvae moved on silk threads to lower leaves on the same corn plant. Sometimes, they moved directly to lower leaves (45.4 ± 5.9%), or, in other cases, they first hung at the end of a thread and swayed like a pendulum, which eventually allowed them to move to a nearby leaf. In the second case, larvae used the silk thread as a balloon to float for various distances to other plants, especially when windy conditions prevailed.

Overall, 1st–3rd instars dispersed through both crawling and ballooning, whereas 4th–6th instars only dispersed by crawling. Ballooning was used primarily by 1st–3rd-instar larvae, but the proportion of the larvae that used ballooning decreased with increased larval age. For example, an average (±SE) ballooning rate of 35.3 ± 7.8% was recorded for 1st instars, followed by 31.7 ± 3.1% for 2nd instars, and 16.6 ± 5.5% for 3rd instars. No ballooning was observed in 4th- to 6th-instar larvae. Given that the rate of ballooning could be related to larval weight, we found that the average weight of 1st instar larvae was 0.04 ± 0.00 mg, compared with 2.76 ± 0.29 mg for 3rd instars, and 243.6 ± 91.9 mg for 6th instars.

The average length of a single silk thread produced by FAW larvae during ballooning was 15.7 ± 1.3 cm (*n* = 60), with the longest length being about 50 cm. When FAW larvae failed to land on a lower leaf, they retracted the silk thread and returned to the original leaf, then crawled or ballooned again. In this way, FAW larvae could disperse by ballooning multiple times. The phenomenon of multiple ballooning attempts was common in FAW 1st instars (24% of 300 larvae tested). 

Crawling is one of the most common dispersal mechanisms used by all FAW larval instars. Newly hatched larvae first lingered on the egg mass and then started to crawl, but in no clear direction, with some moving towards the base of the leaf, some towards the tip, and some moving to the opposite surface of the same leaf. Overall, 23.5 ± 1.4% of larvae crawled towards the base of the leaf. Some larvae that crawled to the leaf tip or blade edge would later disperse by ballooning. For those that reached the corn stalk, about 78% climbed upward and 22% downward (Figure 2). 

### 3.2. Effect of Airflow on Larvae Ballooning

Air flow had a significant impact on the dispersal distance that larvae achieved through ballooning. In the absence of airflow, the farthest distance that the 1st-instar FAW larvae dispersed through ballooning was 71 cm. The highest concentration of larvae (45.4%) was found within 10 cm of the test plant, with fewer larvae found with increasing distance (Figure 3A). The average dispersal distance was 19.33 ± 0.97 cm under wind-free conditions. Such data suggest that many larvae disperse by simply descending with silk thread rather than actually ballooning.

Under the action of airflow, however, the farthest dispersal distance of a 1st-instar larva was 196 cm, and the peak concentration (13.3%) of dispersing larvae was at a distance of 41–50 cm from the test plant. The mean dispersal distance under windy conditions was 75.78 ± 3.50 cm, which differed significantly from wind-free conditions (*p* < 0.000). In comparison to calm conditions, both the distance of the larvae dispersal peak and the farthest distance of the larval dispersal under airflow conditions were much farther (Figure 3B).

The direction of airflow had a strong effect on the direction of ballooning. In the absence of airflow, ballooning dispersal had no obvious directional influence (Figure 4). Larvae were recorded on sticky plates in all four directions (east, south, west, and north) from the test plant with no significant differences among directions (*F* = 2.474, *p* = 0.136). In the case of west-to-east airflow, the number of larvae in the eastern quadrant was significantly higher than in the southern, western, and northern quadrants (*F* = 1352.78, *p* = 0.000).

## 4. Discussion

Larval dispersal of *Spodoptera frugiperda* is an important behavior to understand because it opens a window to comprehend the insect’s intra-plant and inter-plant distribution as well as its expansion at the field level [28,35,37,41]. FAW 1st instars are very small and often suffer mortality rates above 90%, which made it extremely difficult to directly observe their dispersal processes [35,40]. For example, Pannuti et al. [28] only recorded 3.5–6.5% of their tested 1st instars in a field dispersal study, and only about 12% of the larvae were recovered when tested under greenhouse conditions. In our study, 56–83% of the 1st instars were captured on sticky plates and double-sided tape during the various trials, indicating that using sticky plates is a good technique to explore larval dispersal.

FAW larvae dispersal by crawling and ballooning has been previously documented [28,35,37]. Our study has revealed that descending on silk threads is also a mode of dispersal, as we observed about 45% of larvae descending on silk threads. The use of silk threads in dispersal occurs in two ways, namely, descending and ballooning. Descending on a silk thread is an important mechanism by which FAW larvae disperse downward, usually within the same plant. Such behavior allows larvae to rapidly reach lower leaves and thereby locate new food sources and avoid cannibalism [42]. Similarly, larvae of the grapevine moth (*Lobesia botrana*; Tortricidae) move downward mainly by silk descent [43]. Vertical descent on silk threads is likely responsible for the observed distribution pattern of FAW larvae within individual corn plants.

Ballooning and descending on silk are both performed by young larvae. The main similarity between these two dispersal methods is that they both use silk threads, with the difference being that larvae can travel farther by ballooning. We suppose that ballooning or descending with silk is influenced mostly by airflow. That is, ballooning is more common when some degree of airflow occurs, whereas in the absence of airflow, young larvae disperse vertically more often. The larvae of many Lepidoptera families are known to balloon, including the larvae of Cossidae, Geometridae, Lymantriidae, Noctuidae, Psychidae, and Pyralidae [23]. We also observed that, in the absence of airflow, young FAW larvae often moved in a pendulum fashion, apparently trying to land on some nearby object. When not successful, the larvae retracted their silk thread and returned to the original leaf. Similarly, larvae of the grapevine moth use the pendulum swinging technique [43]. Therefore, dispersal by ballooning vs. descending vertically appears to be largely governed by the interactions of the larvae with airflow. 

The impact of airflow on larval dispersal involves at least three factors: direction, distance, and larval density (Figure 3 and Figure 4). As noted above, ballooning is most common when some degree of airflow occurs, and the direction of larval ballooning is influenced by wind direction [36,37]. When the wind direction is highly variable, larval spread of the grapevine moth appears directionless [43]. In our research, the direction of airflow largely determined the direction of larval ballooning. In a study in Nebraska, USA, the prevailing southwest winds were largely responsible for FAW infestations spreading to the north [37]. Ali et al. [41] reported that, with airflow, FAW larvae spread 1–2.4 m, which is somewhat similar to our finding of 0.8–2.0 m. Logically, as the speed of airflow increases, larvae that disperse by ballooning will likely travel farther. In the field, an initial determination of potential larval dispersal locations can be made based on prevailing wind direction for early monitoring and control measures. However, due to the complex microclimates that exist in a corn field, the exact landing points of FAW larvae will be highly random [19,37]. This may explain why the distribution of FAW larvae within a field appears to be random. 

Our greenhouse experiments were conducted with air temperatures in the range of 13.2–25.7 °C and an average relative humidity of 68%. The average temperature was 15.8 °C during the night, and 23.1 °C during the day. Both the average daytime and nighttime temperatures were above the developmental threshold temperature of FAW larvae [16], and therefore we believe that temperature did not adversely affect FAW behavior.

Crawling is the major means of dispersal for FAW larvae and it is performed by all larval instars. FAW 1st through 3rd instars can disperse by crawling or ballooning, while 4th through 6th instars can disperse only by crawling. Body weight is considered an important factor affecting dispersal behavior. Our study found that the average body weight of FAW larvae increased by nearly 100 times from the 1st to the 3rd instars, and then again from the 3rd to the 6th instars. Furthermore, as larvae develop, they acquire more tarsal hooks on their legs and prolegs, which improves their crawling ability [44]. For example, FAW 1st instars are small in size and have few tarsal hooks, whereas the 3rd-4th instar larvae have 10–15 hooks, and the 5th-6th instar larvae have 17–18 hooks [45]. These tarsal hooks provide adhesive properties, which help stabilize the larva’s body and prevent slippage or displacement during movement. 

Our research found that FAW larvae did not show a clear directional preference when crawling on a horizontal plane, as shown by their preference to crawl towards the tips of the leaves as often as the base of the leaves. In contrast, on a vertical plane, when the corn plant is in the vegetative growth stage, most larvae tend to crawl upward along the stalks and congregate on the upper tissues or reproductive organs of the corn plant. These observations are consistent with the study results of Sparks [15] and Volp et al. [46]. FAW larvae prefer to feed on the mesophyll of the upper leaves, and when these leaves more fully develop and open, they present the characteristic pattern of paired holes, which is considered a key feature for identifying FAW damage [6,7,10]. When the corn plant is in the reproductive growth stage, most FAW larvae will occur on the tassels, silks, and corn husks [28]. Feeding by FAW larvae inside the corn husks creates multiple tunnels and holes, which directly affect corn earing and damage level [7,18]. It is reasonable to assume that the choice of suitable food is an important factor affecting the crawling direction of FAW larvae, especially for older larvae. Some studies have reported that the odors of different plant tissues and organs provide cues that the larvae use when locating and moving towards food resources [46,47,48]. Rojas et al. [40] argue that older larvae determine their direction of movement mostly through vision, not scent, and that older larvae appear to show strong directionality when crawling. Dispersal of FAW larvae is driven not only by avoiding mutual competition and cannibalism but also by obtaining food and seeking refuge from predators [21,34]. 

## 5. Conclusions

Ballooning and crawling are regarded as the main measures of FAW larval dispersal. Ballooning only occurs in 1st–3rd-instar larvae and decreases with increasing larval age within these instars. Ballooning allows for larvae to disperse for greater distances. Crawling occurs in all larval instars, but 4th–6th instars can only spread by crawling. FAW larvae will crawl to all above-ground plant parts to feed. Airflow appears to be the main environmental factor affecting the direction and distance of the larval dispersal. When spraying insecticides for prevention or control of FAW in the field, it is necessary not only to focus on the plants with FAW egg masses, but also to spray nearby plants that may have become infested by ballooning larvae.

## Figures and Tables

**Figure 1 insects-14-00488-f001:**
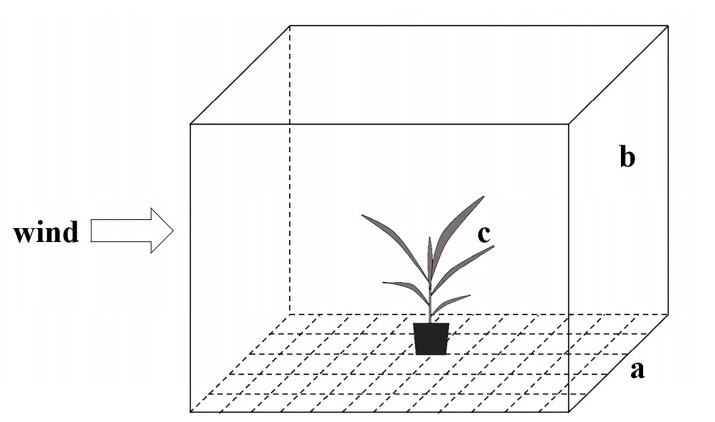
Schematic diagram of the experiment layout, including: (**a**) floor covered with sticky plates (only 60 shown for simplicity); (**b**) wind shelter with four two-m-tall walls, with the two facing the fan being removed when needed; and (**c**) central location of an infested test plant.

**Figure 2 insects-14-00488-f002:**
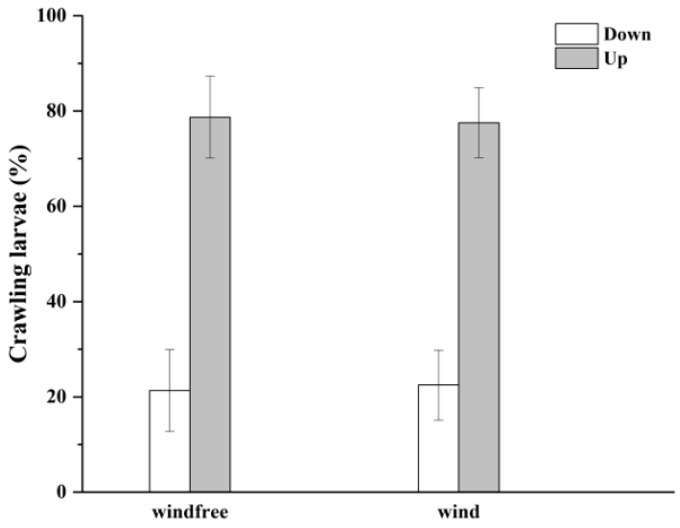
Percent of FAW 1st-instar larvae observed to crawl upward or downward along the stem while experiencing either windy or wind-free conditions after hatching from an egg mass that was placed on a specific leaf of the test plant. The white bar represents the percentage of FAW larvae that crawled downward, while the gray bar represents the percentage of FAW larvae that crawled upward. The bars represent mean ± SE. *p* values are given for the *t*-test of the average percentage of larvae that moved upward or downward while experiencing wind-free or windy conditions. *P*_windfree_ = 0.006, *P*_wind_ = 0.009.

**Figure 3 insects-14-00488-f003:**
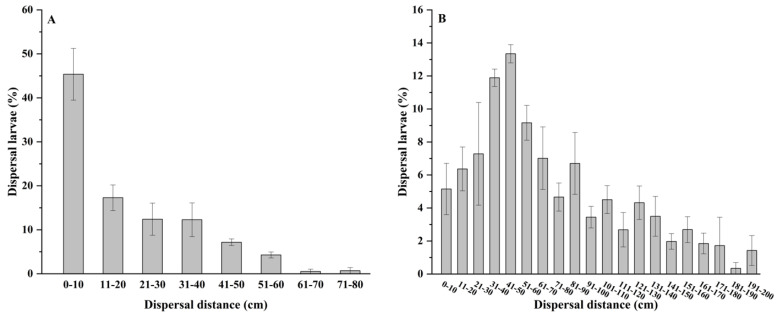
Percent of 1st-instar FAW larvae that dispersed various distances by ballooning from a test plant when experiencing (**A**) no wind or (**B**) some wind produced by a fan.

**Figure 4 insects-14-00488-f004:**
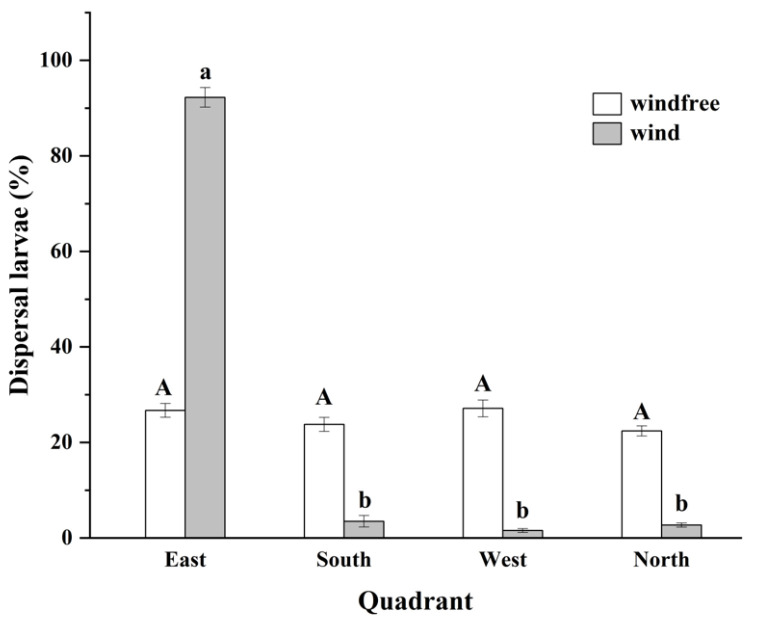
Percent of 1st-instar FAW larvae that dispersed by ballooning and were later collected in one of four directional quadrants relative to the test plant under conditions with no wind or with a mild wind that moved the air west to east. The same uppercase letters indicated no significant differences among the four directional quadrants when tested with no wind. The different lowercase letters indicated there were significant differences among the four directional quadrants when tested with mild wind. *P*_windfree_ = 0.136, *P*_wind_ = 0.000.

## Data Availability

The data presented in this study are available on request from the corresponding authors. The data are not publicly available due to being a subset of an ongoing research project.

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
