# Peer review of "Dispersal Behavior Characters of Spodoptera frugiperda Larvae"

_insects, 2023, doi:10.3390/insects14060488_

Round 1
Reviewer 1 Report
The authors touch upon a very important problem - the protection and preservation of food reserves. The object of investigations – FAW - is important too, because it is distributed in wide range. Due to its high reproductive potential this pest is dangerous and the investigation of its dispersal characteristic of crawling and ballooning in the field and laboratory is both of scientific and practical interest.
Author Response
We appreciate the time that you and the other reviewers spent in reviewing our paper and providing valuable comments, which have led to version. The authors have carefully considered the comments and tried our best to address every one of them. We hope the revised manuscript now meets with your high standards. The authors welcome further constructive comments if any.

Reviewer 2 Report
The Article discusses about the dispersal behavior of larvae of the Spodoptera frugiperda, commonly known as the fall armyworm (FAW). Larval dispersal is a widespread phenomenon in Lepidopteran, and it is considered an important behavior to understand because it opens a window to comprehend the insect's intra-plant and inter-plant distribution as well as its expansion at the field level. The article discusses the factors contributing to the dispersal of FAW larvae, including finding suitable food, reducing intraspecific competition, avoiding predation, and cannibalism. The article also highlights the difficulties in directly observing the dispersal processes of FAW larvae due to their small size and high mortality rates.
The minor and major difficulty with the manuscript is listed below.
The discussion does not provide a broader context for the study of larval dispersal, such as the ecological and economic significance of the phenomenon, the implications for pest management, or the relationship between larval dispersal and other aspects of insect behavior.
Secondly, the paragraphs in discussion lack a clear structure, making it hard to follow the flow of information. The information presented jumps between topics without clear transitions, and some information feels disconnected from the main argument. Further the results obtained by authors needs to be clearly connected with previous knowledge and need to be discussed.

The paragraphs have some minor errors and awkward phrasing, but the overall quality of the English language is good. There are some technical terms and acronyms used in the text, which may be difficult for readers who are not familiar with the subject matter. Additionally, some sentences are quite long and may be challenging to follow. Nevertheless, the paragraphs convey the information effectively and with clarity.
Author Response
We appreciate the time that you and other reviewers spent in reviewing our paper and providing valuable comments, which have led to version. The authors have carefully considered the comments and tried our best to address every one of them. We hope the revised manuscript now meets with your high standards. The authors welcome further constructive comments if any.
Below we provide a point-by point responses to the review comments. Since the line numbers of the revised version and the original version are inconsistent, we first give the line numbers from the revised version followed by the original version in parentheses.

Reviewer 3 Report
This paper presents studies on the dispersal of fall armyworm (FAW) larvae by crawling and ballooning. The paper reads well, and the authors seem to know the subject well. I have made comments, suggestions, and some minor editorial changes in the body of the paper using comments tools. However, I have a few issues with structure and form of the paper, which is why I’m suggesting major revisions.
First, the Introduction provides very little information on what previous work has been conducted on dispersal of FAW. Because FAW is a well-known and well-studied pest of several crops across the globe, there should be an extensive body of literature on previous studies similar to this one. This is especially true with FAW, as (at least in the U.S.) this pest species migrates from southern latitudes northward each growing season, making dispersal mechanisms a primary concern in management. However, most of the literature presented is in the Discussion section, and much of it in the Discussion is general information that could (should) be moved to the Introduction and tailored to the current study. It is important to present this information in the Introduction, rather than saving it for the Discussion, to 1) demonstrate an honest effort of the authors to know the subject matter, but more importantly 2) to distinguish the authors studies from previous work. If the reader is not presented with a summary of previous similar work, there is no metric to determine the significance of the current study being presented. With a pest such as FAW that has been the subject of thousands of studies, it is important to distinguish the significance of the work being presented from previous work. As the paper currently reads, I’m not sure if the findings greatly contribute to the body of knowledge of FAW dispersal behavior or not.
Second, but related to the development of the Introduction, the objectives presented are very vague, especially for a species like FAW. If this paper were studying a less well-known species, I could understand the generality of the objectives. However, I think the authors need to rethink and reword especially the first objective to better reflect what they did/what they were trying to accomplish.
Third, there is a bit of disconnect between some studies presented in the Methods and the presentation of data for these studies in the Results. For example, 1st, 3rd, and 6th larval instars were weighed during the rearing process to analyze the relationship of larval weight to ballooning. However, there is only one sentence in the Results that relates to this, and it just presents the mean weights of each instar class. There is no analysis involved linking, for instance, larval weight and dispersal distance of ballooning larvae or any other dispersal factor or behavior. Another example is the study where FAW eggs and larvae were monitored in an open corn field. There are only two very general summary sentences related to the corn field study in the Results. No analyses were conducted on any data associated with this study, nor were any comparisons made between observations in the field and in the greenhouse. For both these examples, it would almost be better to remove these items from the Methods and Results, rather than include them in their current state. I would rather see the authors develop the results of these studies and relate them more strongly to the more controlled greenhouse study.
Fourth, the data analysis section in the Methods needs to be developed a bit more. There is no description of the homogeneity test or the data transformation. Also, if additional studies are analyzed (as suggested above), the authors need to include the methods and techniques used to analyze those studies.

Author Response
We appreciate the time that you and the reviewers spent in reviewing our paper and providing valuable comments, which have led to version. The authors have carefully considered the comments and tried our best to address every one of them. We hope the revised manuscript now meets with your high standards. The authors welcome further constructive comments if any.
We provided a point-by point responses to the review comments in the attachment. Since the line numbers of the revised version and the original version are inconsistent, we first give the line numbers from the revised version followed by the original version in parentheses.

Round 2
Reviewer 3 Report
I would like to thank the authors for addressing the previous comments so thoroughly. I think the paper reads better and its significance is more clearly seen. I have made some comments on the new version of the paper, but most are minor. The authors did leave out Figure 2 in this version, but I think it was an oversight and not a permanent omission. I also suggested reconsidering their final statement in the Conclusions. Upon addressing my current minor comments, I feel this paper is ready for publication.

Author Response
Dear Editors and Reviewers,
We appreciate the time that you and the reviewers spent in reviewing our paper and providing valuable comments, which have led to version. The authors have carefully considered the comments and tried our best to address every one of them. We hope the revised manuscript now meets with your high standards. The authors welcome further constructive comments if any.
Below we provide a point-by -point responses to the review comments.
Sincerely yours,
Yong-Ping Li & Co-authors
